# Ground Force Precision Calibration Method for Customized Piezoresistance Sensing Flexible Force Measurement Mat

**DOI:** 10.3390/s24072363

**Published:** 2024-04-08

**Authors:** Jeong-Woo Seo, Hyeonjong Kim, Jaeuk U. Kim, Jun-Hyeong Do, Junghyuk Ko

**Affiliations:** 1Digital Health Research Division, Korea Institute of Oriental Medicine, Daejeon 34504, Republic of Korea; jwseo02@kiom.re.kr (J.-W.S.); jaeukkim@kiom.re.kr (J.U.K.); jhdo@kiom.re.kr (J.-H.D.); 2Department of Mechanical and Aerospace Engineering, University of California, Los Angeles, CA 90095, USA; koriente@g.kmou.ac.kr; 3Korean Convergence Medicine, University of Science and Technology, Daejeon 34054, Republic of Korea; 4Open XR Platform Convergence Research Center, National Research Council of Science and Technology, Daejeon 34141, Republic of Korea; 5Division of Mechanical Engineering, (National) Korea Maritime and Ocean University, Busan 49112, Republic of Korea

**Keywords:** piezoresistance sensor, force plate, center of pressure, deep neural network, calibration

## Abstract

A force plate is mainly used in biomechanics; it aims to measure the ground reaction force in a person’s walking or standing position. In this study, a large-area force mat of the piezoresistance sensing type was developed, and a deep-learning-based weight measurement calibration method was applied to solve the problem in which measurements are not normalized because of physical limitations in hardware and signal processing. The test set was composed of the values measured at each point by weight and the value of the center of the pressure variable, and the measured value was predicted using a deep neural network (DNN) regression model. The calibration verification results show that the average weight errors range from a minimum of 0.06% to a maximum of 3.334%. This is simpler than the previous method, which directly measures the ratio of the resistance value to the measured weight of each sensor and derives an equation.

## 1. Introduction

Various principles and types of sensors are used to measure the functions of kinetic characteristics [1]. In particular, the force plate, which is primarily used in biomechanics, aims to measure the ground reaction force. It is used to measure the dynamic reaction force, such as in walking, and the static reaction force, as in standing, and to measure the direction and magnitude of the force movement.

Most commercial force plates have load cells that may contain piezoelectric elements, strain gauges, or beam-load cells [2,3,4]. There are two main types of force plate. The first is a general force plate. These are the data measured in the load cell inserted at the four rectangular plate edges, and the force in the three-axis direction is summed, measured, and presented as a force vector value in consideration of the direction of the force [5]. The other is the central-support-type force plate. The ground reaction force is calculated from the moment measured at the pillar fixed at the bottom of the plate [6]. Thus, the voltage measured by the load cell is proportional to the applied force. Advantageously, the force plate enables relatively accurate measurements and is fabricated from a hard material with a flat surface that facilitates measuring the movement. Another type of kinetic measurement device is a force sensor.

Force sensors are used to determine the value of the force measured in the walking or standing postures or the shape of the positional change of the sensed force, which consists of an array of force sensors (FSRs) or piezoelectric sensors on the surface. The dynamic perspective of the human physique is treated as a ridged body. Force measurement data for kinetic analyses require precise and sophisticated values. The value of the force is used as information about the overall external force to return the joint torque to the inverse dynamics [7]. The calculated joint torque is also used to predict muscle activity along with the acceleration and length values of the limb [8]. Accuracy must be guaranteed for use in medical and healthcare diagnosis [9]. Therefore, the commercial force plate for motion analysis converts the value of the force applied to the load cell into an electrical signal, precisely calculates and provides the value of the force corresponding to the voltage value, and indicates the error level [10].

Several studies have been conducted to confirm the accuracy of measurement values and calibrate the values of force measurement systems using diverse sensors, such as load cells and FSRs, which have been developed for use in research as well as commercialized force measurement systems. Weizman identified a novel method for validating the force and center of pressure (CoP) obtained from pressure-measuring insoles using commercially available equipment [11]. Faber proposed a six-degree-of-freedom force/torque sensor (FTsensor) using a precalibrated force plate (FP) as a measurement reference method for device calibration [12]. Bobbert evaluated the accuracy of commonly used plates and proposed a calibration algorithm to improve it [13]. These studies measured the values at each position of the force plate and derived a conversion formula from the measured values. However, in FSR and piezoelectric-type measurement devices composed of multiple sensors, the pattern of each sensor value may be irregular depending on the data communication speed and processing method.

In this study, we present a measurement and correction method for calibrating the force of a customized capacitance- and resistance-sensing-type flexible force measurement mat developed in-house.

## 2. Materials and Methods

### 2.1. Introduction of Flexible Force Measurement Mat

This sensor is of the piezoresistance sensing type, and the size of one flexible sensor cell is 21 × 36 × 2.5 mm. A total of 3264 were placed in a 1530 × 1820 mm area comprising 68 rows and 48 columns [14]. A piezoresistance sensor is made from semiconductor material. The resistance of this varies greatly when the sensor is compressed or stretched [14]. A thin electrode connects each sensor, and there is a large area of polyethylene coated on both sides for protection. The measured data are obtained following the order of each column and row from the upper PCB and transmitted to a connected PC. The sensor unit being measured and the upper PCB area for data acquisition and processing are connected to a socket jack connector port in the form of an electrode port (Figure 1). The power of the system is DC 5.0 V, 2.0 A using a micro USB, and the measured data are obtained at a sampling frequency of 20 Hz through a dedicated PCB, with a force range of 10 kg to 100 kg. The serial port is connected by a USB, and the measured pressure distribution of the flexible force mat connected to the console PC is given as an image. A 15 mm thick polyvinyl chloride safety mat is adhered to the top surface of the force mat to reduce slipping and enhance safety (Figure 2).

### 2.2. Verifying Flexible Force Mat Acquisition Data

Considering that this system is used to measure the standing posture or walking ability of the human body, the center of pressure (CoP) was calculated from the value measured when supported with both feet. The CoP was calculated as (location information of each cell × measured value)/total pressure value of each foot, and the formula is shown in Figure 3 and Equations (1)–(3) [15].
(1)$${CoP}_{row}=\frac{\sum {Row}_{n}\times Value}{\sum Value}$$
(2)$${CoP}_{column}=\frac{\sum {Column}_{n}\times Value}{\sum Value}$$
(3)$${CoP}_{value}=\frac{\sum Value}{n}$$

The CoP was calculated from the sum of values measured from the sensor in the row position, and the change in the sensor value compared with the same weight was confirmed by calculating the number of columnar measured CoPs from the center {1,34} position to the {48,34} position. The sensor row and column numbering defined the opposite lower left side as position {1,1}; with the PCB module mounted at the top, the columns increased to {48} and the rows increased to {68}.

### 2.3. Development an Algorithm for Flexible Force Mat Weight Estimation

A deep neural network-based predictive deep learning regression model was developed to predict the value measured by the sensor and output a kilogram value. To construct a dataset, it is difficult to measure the value of each sensor, which is small; therefore, the value of 463 points was measured with 63 intersections and 400 random points along the rows and columns to match the average foot size, and the features were calculated.

There are a total of 6 features used in model learning. The “Sensor value” is the total value of data in activated cells. The “No. Cell” means the number of sensors activated during weighting, and “CoP row” and “CoP column” are values calculated using formulas 1 to 3. The “Row Gap” and “Column Gap” refer to the width that is activated in the top, bottom, left, and right directions (Table 1).

The training data were the CoPs and location-related features that could be measured and calculated from the ground reaction force. First, using 20, 40, and 60 kg weights, pressure distribution information at 20 Hz sampling frequencies was collected at a total of 63 locations in rows 7 and 9 within an array of 68 × 48, and the weights’ data were randomly collected at 400 locations to secure additional data for training. Six training feature sets were acquired from one array and calculated to construct a set of learning data for each array (Table 1). For example, if the data were obtained by applying 60 kg for approximately 30 s at position A, the frequency would be 20 Hz, resulting in a total of 600 independent matrices. Thus, 600 lines of data consisting of a set of features were generated from a single measurement. These 600 matrices differ only slightly from each other, making it difficult to find significant differences; however, this method for data expansion is more reliable than generating the data virtually. The training dataset’s 833,400 lines were completed by accumulating data extracted from hundreds of matrices in each experiment (Figure 4). The entire dataset was constructed, and measurements exceeding 3 times the standard deviation of the repeated measurements were defined as outliers and excluded.

Deep neural network (DNN) regression, a deep neural network, was used to develop the predictive model. Respectively, the dataset, test, and validation sets were randomly divided into 60%, 20%, and 20% groups. The optimizer and loss function, respectively, were adaptive momentum estimation (ADAM) and mean squared error (MSE). ADAM integrates the momentum algorithm and the RMSProgram [16]. The input shape of the hidden layer of the learning model was six, consisting of 150 nodes. The hyperparameters were set to 500 epochs and a batch size of 100 (Figure 5).

To verify the developed regression model’s accuracy, the mean absolute error (MAE), mean squared error (MSE), and R-squared (R^2^) values were calculated using 20% of the test set data as input values (Equations (4)–(6)) [17].
(4)$$MAE(\mathrm{Mean}\;\mathrm{Absolute}\;\mathrm{Error})=\frac{\sum \left|y-\hat{y}\right|}{n}$$
(5)$$MSE(\mathrm{Mean}\;\mathrm{Squared}\;\mathrm{Error})=\frac{\sum_{i=1}^{n}{\left(y-\hat{y}\right)}^{2}}{n}$$
(6)$$R^{2}(R-Squared)=1-\frac{SSE}{SST},\;SSE=MSE,\;SST=\frac{1}{n}\sum_{i=1}^{n}{\left(y^{i}-\bar{y}\right)}^{2}$$

### 2.4. Validation Using Real Data

To confirm the accuracy of the deep learning model developed using the data measured using the weight of the disk, an additional experiment was conducted using a standard weight similar in size to a square foot-shape. Measurements were taken at a total of 9 points arranged by row and column. Based on the measurement center point sensor, the 9 coordinates were #1.{13,41}, #2.{13,25}, #3.{13,11}, #4.{34,41}, #5.{34,25}, #6.{34,11}, #7.{51,41}, #8.{51,25}, and #9.{51,11}.

The purpose of this study was to confirm whether accurate weight estimation is possible, even with weights of different shapes. The weight (kg) was estimated by placing rectangular weights of 20, 40, and 60 kg at nine positions on the force mat and inputting the measured data into the customized deep learning model. See Figure 6.

## 3. Results

### 3.1. Force Mat Weight Sensor Value

Figure 7 and Table 2 show the results of the values measured using 15 kg weights placed in rows spanning the 4th to 45th columns, based on sensor positions 12, 20, 28, 35, 43, 50, and 58. Based on each row, the value of the column tended to be 0.9 or higher when checked by the regression equation; however, when compared to the average value, there was a variation in the sensor value for each location.

The value of each location was measured, and the average value for each row position was calculated. As a result, it was confirmed that the closer to the top where each location PCB module was inserted and the closer to the center than to the left and right, the higher the value (Figure 8).

### 3.2. Results of Validating the Deep Learning Model on the Test Set

The model used data that were not used for learning, at a rate of 20%, to confirm its performance [17,18]. The MSE, which is the loss function between the predicted value and the actual target value, was 1.078, the MAE, which is the regression index, was 3.330, and R^2^, which is the coefficient of dimension, was 0.985, confirming high prediction accuracy.

### 3.3. Results of Validating the Deep Learning Model on Real Data

To verify actual usability, the results of the estimated kg value using calibrated weights of 20, 40, and 60 kg as input values are shown in Table 3. As shown in Figure 4, the average error of the regression model results calculated from the values measured at a total of nine locations was confirmed to be 0.06% for 20 kg, −2.14% for 40 kg, and 3.34% for 60 kg.

## 4. Discussion

In this study, the specifications of the pressure sensor applied to the custom-made force mat were the same, but the measurement values for the same weight were different depending on the power for transmitting and receiving data and the limitations of the data processing module. Therefore, the sensor had to be calibrated [19]. Initially, a simple conversion equation was derived by measuring all sensors, and this was commonly used for calibration. However, this study proposed a method of calibration by developing a regression model based on deep learning to estimate weight using a simplified measurement method [20,21]. To develop a calibration model, the characteristics of the trends in the measured values must first be checked for each sensor. The first result of this study confirmed that there was no tendency in the measured value at the designated location, which means that there is a limit to performing calibration using a simple linear correction formula. Looking at the tendency of each value, it was confirmed that it changed to a nonlinear value, and the row position decreased closer to the PCB module at the top of the data-receiving unit and decreased as the distance increased. The column position tended to increase as it approached the center and decreased as it moved farther to the side. Because the power differs depending on the order of transmission and reception of the data and the distance at which the power is applied and transmitted, it is thought that there will be a difference in the measured value [22]. Various studies have been conducted to correct errors caused by factors such as the order of transmission, reception, and voltage differences [23,24,25,26]. A previous study measured and calculated the hysteresis for the value of the force applied in calibrating a force-sensing resistor, and a digital filter was designed to perform individual matching of the measured value of the sensor [26]. This is believed to be heavily influenced by face-to-face problems and hardware specifications for processing the measured data. Measurement calibration is required to solve this problem, and it is difficult to create a regression equation owing to the inconvenience of direct measurement and the low tendency of measurement values. Therefore, deep-learning-based calibration methods have recently been developed and applied. In related prior research, a method for setting the multiaxis force as a raw reference dataset and estimating and calibrating the value using a deep natural network was proposed to solve the coupling effects and nonlinearity problem caused by the force torque sensors used in robotic arms [26]. The learning features used to develop a deep learning model to predict measurement weight were the sum value of data in the activated cells, the number of activated cells, the row and column points of the CoP activated area, and the activated area in rows and columns. This system is meant to calculate the CoP based on measuring the ground reaction. Rather than calibrating individual sensor values, the dataset was set and considered using the weights of foot-sized objects. In addition, various correlated features were set to estimate the weights so that more accurate predictive models could be developed. In the deep learning model’s validation results that were verified using the test set, the MSE which is the loss function between the predicted and actual target values was 1.078. The MSE value indicates the similarity between the predicted and actual values, and the closer it is to zero, the better the fit. This confirms that the estimated evaluation of the model is possible because it shows a relatively low value [27]. The MAE was 3.330, and the R-squared value was 0.985, which is close to 1 [28]. This shows that the estimation result for the applied input value is excellent. This system will be commercialized in the future and the lowest (0.06%) and highest (3.34%) error rates were found in the results calculated using this model. The weight used for learning was circular. However, the standard weight for verification was rectangular. The error tended to increase as the applied weight increased. This is expected to be an error caused by the characteristics of the sensor, and the data loss characteristics of this sensor must be checked more precisely; this would be reflected in the calibration.

## 5. Conclusions

Various measurement errors occur in the piezoresistance sensing type large area force plate used to measure ground reaction force. This is caused by the processing delay time of the measured data and the voltage difference applied for measurement, so weight measurement calibration work for each sensor is required. In this study, a CoP-based deep learning regression model was developed with more simplified measurements to solve the difficulty of developing repetitive measurements and individual calibration correction equations for all sensors, and accuracy was secured. It is not suitable as a method for precise measurement, but the possibility of use was confirmed as a method that can estimate the CoP of a person’s walking or standing posture and estimate quantitative weight. In future research, it is necessary to develop more diverse deep learning models to supplement precision.

## Figures and Tables

**Figure 1 sensors-24-02363-f001:**
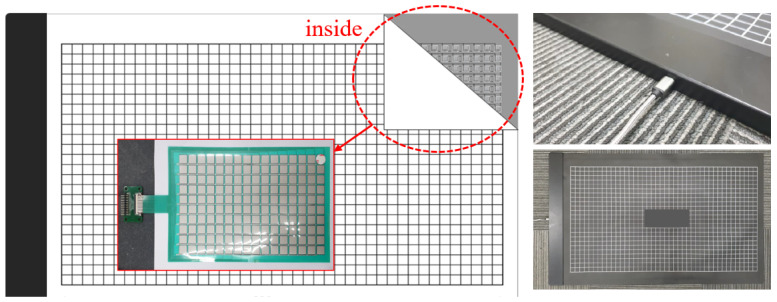
Flexible force measurement mat with sensor and electrode (without safety mat cover).

**Figure 2 sensors-24-02363-f002:**
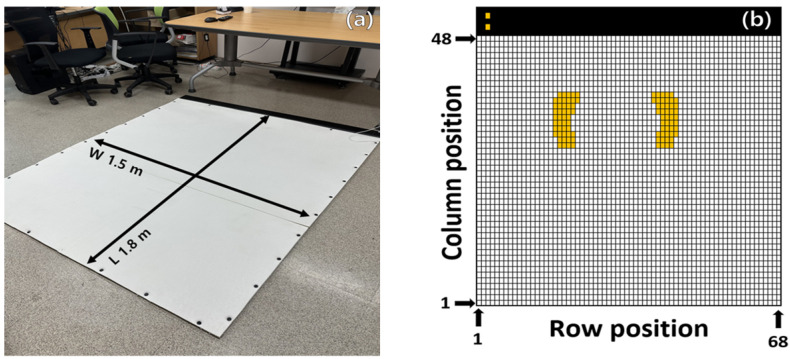
Flexible force measurement mat with safety mat cover (**a**), array number (**b**).

**Figure 3 sensors-24-02363-f003:**
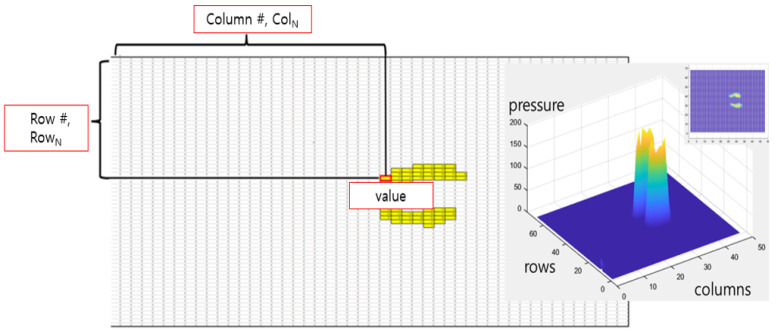
Flexible force measurement mat acquisition data {row, column} and example of foot pressure.

**Figure 4 sensors-24-02363-f004:**
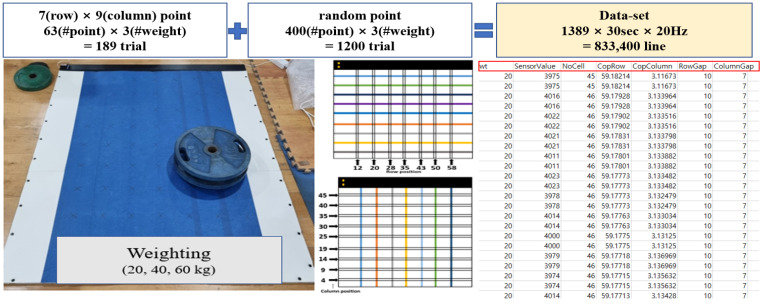
Preparing the training dataset.

**Figure 5 sensors-24-02363-f005:**
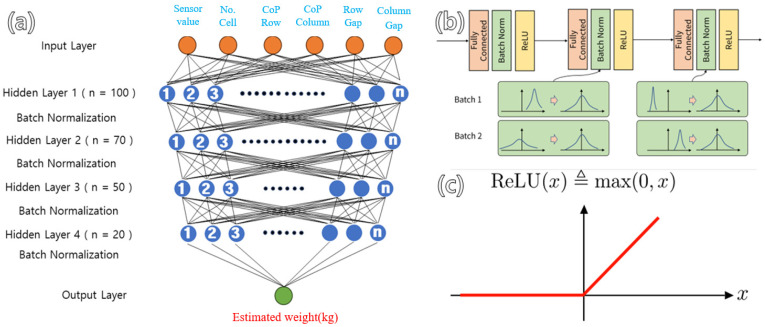
(**a**) Architecture of DNN; (**b**) Optimizer; (**c**) Rectified Linear Unit (gradient function).

**Figure 6 sensors-24-02363-f006:**
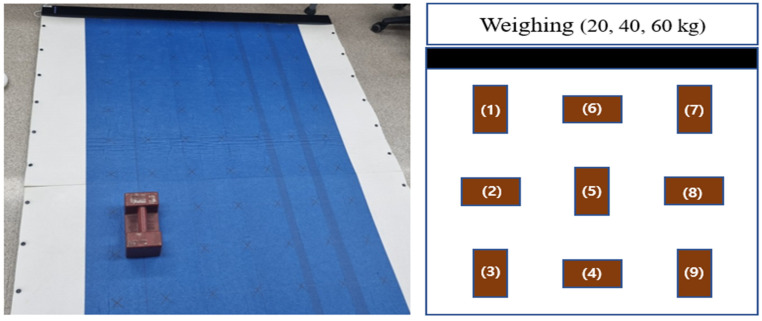
Position of real acquisition data.

**Figure 7 sensors-24-02363-f007:**
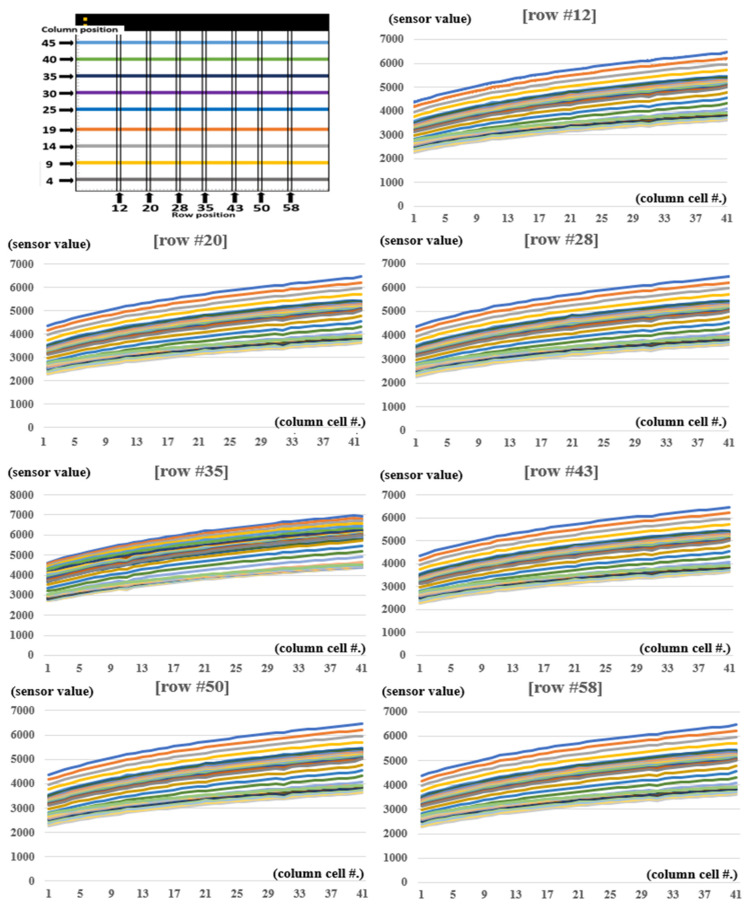
Results of sensor value checked (#: number).

**Figure 8 sensors-24-02363-f008:**
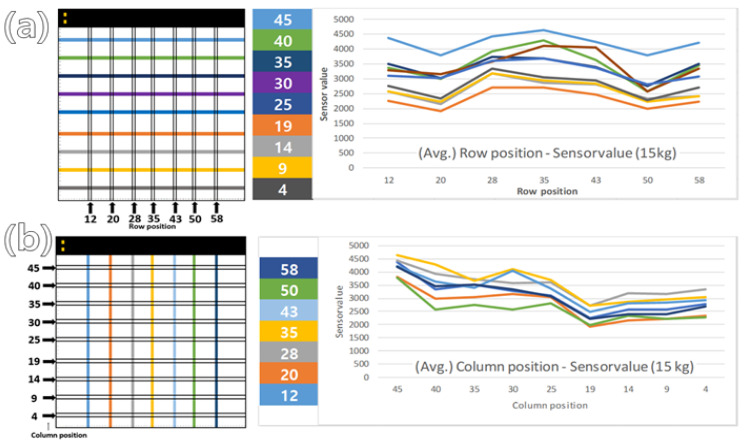
Results of average of (**a**) row and (**b**) column position sensor value.

**Table 1 sensors-24-02363-t001:** Feature for DNN algorithm.

Feature	Description
Sensor value	Total values of data in activated cells
No. Cell	Number of activated cells
CoP Row	Row point of the CoP-activated area
CoP Column	Column point of the CoP-activated area
Row Gap	Row width of activated area
Column Gap	Column width of activated area

**Table 2 sensors-24-02363-t002:** Average of sensor values: 15 kg weighting.

Row Column	12	20	28	35	43	50	58
45	4361.81	3801.10	4432.26	4626.48	4231.57	3788.62	4208.67
40	3353.14	2993.63	3920.44	4278.25	3627.51	2576.39	3447.71
35	3505.48	3031.93	3726.73	3672.93	3399.90	2749.87	3505.99
30	3281.29	3164.10	3564.50	4103.60	4061.09	2572.16	3335.32
25	3113.13	3034.10	3612.79	3688.41	3364.83	2817.65	3064.23
19	2263.09	1910.09	2707.31	2714.86	2477.88	1980.80	2218.33
14	2570.04	2159.12	3191.69	2860.07	2800.15	2345.33	2409.73
9	2562.44	2233.04	3170.10	2948.55	2828.51	2223.13	2409.29
4	2766.91	2334.68	3327.82	3042.83	2931.93	2289.39	2692.63
Average	3086.37	2740.20	3517.07	3548.44	3302.60	2593.70	3032.43
SD (±)	635.73	611.28	496.67	691.22	597.08	520.87	654.53

SD: standard deviation.

**Table 3 sensors-24-02363-t003:** Results of the deep learning model’s estimation error on real data.

Weighing Point	20 kg	Error (%)	40 kg	Error (%)	60 kg	Error (%)
1	19.57	2.15	40.53	−1.33	57.46	4.23
2	19.80	1.00	43.74	−9.35	54.19	9.68
3	20.49	−2.45	42.69	−6.72	60.08	−0.13
4	19.18	4.10	40.89	−2.22	60.51	−0.85
5	20.26	−1.30	39.96	0.10	58.38	2.70
6	20.44	−2.20	42.34	−5.85	60.84	−1.40
7	19.91	0.45	39.41	1.48	55.33	7.78
8	20.33	−1.65	38.96	2.60	54.96	8.40
9	19.92	0.40	39.19	2.03	60.19	−0.32
Average	19.99	0.06	40.86	−2.14	57.99	3.34
SD (±)	0.44	2.18	1.70	4.26	2.62	4.36

SD: standard deviation.

## Data Availability

The data presented in this study are available upon request from the corresponding author.

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
