# Peer review of "Ground Force Precision Calibration Method for Customized Piezoresistance Sensing Flexible Force Measurement Mat"

_sensors, 2024, doi:10.3390/s24072363_

Round 1
Reviewer 1 Report
Comments and Suggestions for Authors
The paper analyses the development and testing of a force mat. The study introduces an innovative calibration approach for a flexible force measurement mat, integrating piezo-resistive sensor technology. It is well written. It offers a novel tool for researchers and practitioners in sports science and rehabilitation.
Below suggestion:
- Introduce new keywords instead of "calibration method" and "piezo-resistance sensing" because they are already in the title.
- Ensure uniformity in the references, as only the first reference includes the DOI.
Introduction
It would be useful to provide more information about the sensor and electrode used in the construction of the mat. If the construction of the mat is to be reproduced, it is not clear whether all types of sensors and electrodes can be used or whether specific ones are required.
The text describes the arrangement and connection of the sensors but does not detail how they are physically incorporated into the mat. For example, how are the sensors and electrodes attached to the mat substrate to maintain flexibility and ensure an adequate response to mechanical stimulation?
In addition, it may be necessary to discuss why the polythene coating was chosen and whether this material is suitable for the sensors, especially if they are vulnerable to interference or damage from environmental factors such as humidity, mechanical stress or electronic noise.
Material and Methods
No pre-processing of the data is described before applying deep learning, hasn't it been done?
In line 102 needs to put the coordinates of the position in curly brackets, as for the previous lines.
The use of a regression model based on a deep neural network to predict sensor values is a notable aspect. However, more information on the network's configuration, including its architecture and training process, would enhance understanding of the model's effectiveness.
Figure 3 comprises three images, with the last one being less visible.
The section describing the methodology for validating the model with real data could be strengthened by providing a more detailed description of the configuration and experimental process. For instance, although figure 4 shows the nine study positions, it would be beneficial to mention the coordinates corresponding to each position in the text.
Results
On line 175, Figure 4 is referenced, but this figure pertains to the validation tests for the nine positions, not the initial collection with the 15 kg.
Line 178, use capital F.
Suggest introducing Table 2 before Figure 6.
In figure 6, on the legend, identify a) and b).
L206, considering Table 3 data the average error for the 20 kg weight is 0.06%, not 0.4% as stated.
Discussion
A direct comparison with other calibration methods or force measurement would be interesting as well as identifying advantages of the mat.
Conclusions
Consider the interest of mentioning potential real-world applications of the force measurement mat.
Author Response
Response to Reviewer 1 Comments
Thank you for your review comments on this manuscript. I think it helped a lot to improve the completeness of this manuscript by your review comment. I actively responded to the lack of content given by the reviewer and tried to apply it to the manuscript. Once again, thank you for reviewing this manuscript, and please check the following for responses to comments.
Point 1: Introduce new keywords instead of "calibration method" and "piezo-resistance sensing" because they are already in the title.
Response 1: According to the comments, change “keywords” to Piezo-resistance sensor; force plate; center of pressure; deep neural network; calibration; It has been modified. thank you.
Point 2: Ensure uniformity in the references, as only the first reference includes the DOI.
Response 2: DOIs was removed to all references to ensure consistency. If publication is confirmed in the future, we will add or modify it in galley proof. Thank you.
Point 3 : It would be useful to provide more information about the sensor and electrode used in the construction of the mat. If the construction of the mat is to be reproduced, it is not clear whether all types of sensors and electrodes can be used or whether specific ones are required.
Response 3: Each sensor cell is a sensor that applies a piezo-resistance type pressure measurement method. It is difficult to explain more specific sensor specifications due to limitations from a commercial copyright perspective. However, please understand that the sensor image has been added to figure 1. Thank you.
Point 4 : The text describes the arrangement and connection of the sensors but does not detail how they are physically incorporated into the mat. For example, how are the sensors and electrodes attached to the mat substrate to maintain flexibility and ensure an adequate response to mechanical stimulation?
Response 4: The array of sensors is physically attached using ports applied to the PCB where data is integrated. Because the measurement sensor unit and the upper black part of the board are not flexible, there is a risk of separation. According to your suggested opinion, “The sensor unit being measured and the upper PCB area for data acquisition and processing are connected to a socket jack connector port in the form of an electrode port.” has been added (lines 79-81). Thank you.
Point 5 : No pre-processing of the data is described before applying deep learning, hasn't it been done?
Response 5: In general, data preprocessing for deep learning means 1) feature engineering, 2) removal of missing values ​​and outliers, and 3) separation of data for learning and evaluation. Which of the three do you mean by no preprocessing? 1) Feature engineering explained how to calculate and construct a data-set like Table 1 and Feature 3 used as features in DNN. 2) For removal of missing values ​​and outliers, measured values ​​exceeding 3 times the standard deviation among the measured values ​​were defined as outliers and excluded, and were added to line 136-137 of “2.3 Development an algorithm for flexible force mat weight estimation”. 3) Separation of data for learning and evaluation has already been written in line 155 as “the dataset, test, and validation sets were randomly divided into 60%, 20%, and 20% groups.” If this is not your intention, please provide specific explanation.
Point 6 : In line 102 needs to put the coordinates of the position in curly brackets, as for the previous lines.
Response 6: The position value corresponding to the matrix has been modified to curly baracket “{ }”.
Point 7 : The use of a regression model based on a deep neural network to predict sensor values is a notable aspect. However, more information on the network's configuration, including its architecture and training process, would enhance understanding of the model's effectiveness
Response 7 : I highly respect your opinion. A figure (figure 5, line162) on network configuration has been added. thank you.
Point 8 : Figure 3 comprises three images, with the last one being less visible
Response 8 : I highly respect your opinion. A figure 4 (revision before; figure3), the last image has been replaced with a higher resolution image. thank you.
Point 9 : The section describing the methodology for validating the model with real data could be strengthened by providing a more detailed description of the configuration and experimental process. For instance, although figure 4 shows the nine study positions, it would be beneficial to mention the coordinates corresponding to each position in the text.
Response 9 : Based on your opinion, we have presented 9 measurement coordinates (lines 173-176). thank you.
Point 10 : On line 175, Figure 4 is referenced, but this figure pertains to the validation tests for the nine positions, not the initial collection with the 15 kg.
Response 10 : There was an error in the way it was expressed. The reference to “figure 4” in Line 175 (revision before version) has been deleted. thank you.
Point 11 : Line 178, use capital F. Suggest introducing Table 2 before Figure 6. In figure 6, on the legend, identify a) and b).
Response 11 : It was modified to capital F, and the positions of figure 7(revision before; figure 6) and table 2 were changed. And, I added a) and b) in figure 6 to the caption so they can be identified. Thank you.
Point 12 : L206, considering Table 3 data the average error for the 20 kg weight is 0.06%, not 0.4% as stated.
Response 12 : “0.4” in line 225 has been modified to “0.06”. Thank you for your careful confirmation.
Point 13 : A direct comparison with other calibration methods or force measurement would be interesting as well as identifying advantages of the mat
Response 13 : I looked for references on various calibration methods, but it turns out that there is a lack of prior research at a level that allows for direct comparison. I will conduct a comparative study based on ground truth CoP in future research through a more careful search of previous papers. Thank you for your very helpful comments.
Point 14 : Consider the interest of mentioning potential real-world applications of the force measurement mat.
Response 14 : In fact, as a follow-up study, we are planning a study to confirm accuracy based on dynamic data from real people. Once the results are obtained, I will report on the follow-up study. Thank you for your good comments.

Reviewer 2 Report
Comments and Suggestions for Authors
In this work, the authors introduce a large-area force mat basing on piezo-resistance sensors, which can be used to estimate the center of pressure (CoP) of a person’s walking or standing posture and quantitative weight. A deep-learning-based weight measurement calibration method is developed to solve the difficulty of developing repetitive measurements and individual calibration correction equations for all sensors. A series of experiments are carried out to validate the method. However, the weight estimation method for the force mat is not introduced clearly enough. Some results of experiments should be explained in detail. And several experiments should be complemented. So, I suggest that the authors could improve the quality of the manuscript by correcting these issues below properly.
1. The author should also focus on the sensor technologies and its error factors, illustrating the working mechanisms, characteristics, and effects of the scheme.
2. I suggest that the top and bottom of Figure 1 a&b should be aligned respectively, and a certain distance should be kept between the two figures.
3. Headings for Figure1 a&b should be added to the title of Figure 1, as well as the other figures in the manuscript.
4. Formulas should be listed separately, rather than in the diagram (See Figure 2). The variables in the formulas should be defined or explained.
5. The weight prediction method should be explained more detailed and clear, in the first paragraph of section ‘2.3 Development an algorithm for flexible force mat weight estimation‘.
6. The Description is unclear in Table 1. It needs further explanation.
7. The curves in Figure 5 should be explained in details.
8. I suggest authors introduce and cite some excellent work about flexible sensor system, such as Cell Reports Physical Science 2023, 4:101191.
9. Since the authors claim that the force pad can estimate the CoP of a person’s walking or standing posture and quantitative weight, it’s better to complement experiments on human bodies.
Comments on the Quality of English Languageminor revision
Author Response
Response to Reviewer 2 Comments
Thank you for your review comments on this manuscript. I think it helped a lot to improve the completeness of this manuscript by your review comment. I actively responded to the lack of content given by the reviewer and tried to apply it to the manuscrip. Once again, thank you for reviewing this manuscript, and please check the following for responses to comments.
Point 1: The author should also focus on the sensor technologies and its error factors, illustrating the working mechanisms, characteristics, and effects of the scheme.
Response 1: I agree with the opinion you presented. The cause of the error is related to the data communication of the sensor. This is because a large number of matrix-type data is input at once, but there is a limit to the hardware performance that can be processed. I suggested this in the introduction, and although there are calibration methods through individual measurements, the purpose was to check the possibility of calibration in a simpler method using deep learning techniques due to the limitations of repeated measurements. We sympathize with the presentation method you suggested, and please understand that we have made efforts to reflect it as much as possible. Thank you.
Point 2: I suggest that the top and bottom of Figure 1 a&b should be aligned respectively, and a certain distance should be kept between the two figures
Response 2: As you suggested, I added more gap between (a) and (b). Please understand that the top and bottom layout has been arranged as before due to limitations on the paper pages. thank you
Point 3 : Headings for Figure1 a&b should be added to the title of Figure 1, as well as the other figures in the manuscript.
Response 3: Added the caption of figure1 (figure2 after modification) to distinguish a) and b). thank you.
Point 4 : Formulas should be listed separately, rather than in the diagram (See Figure 2). The variables in the formulas should be defined or explained.
Response 4: Formulas and figures are separated as shown on page.3. thank you
Point 5 : The weight prediction method should be explained more detailed and clear, in the first paragraph of section ‘2.3 Development an algorithm for flexible force mat weight estimation‘.
Response 5: I agree with you. Figure 5 was added to page 5 because the explanation of the architecture of the DNN model was thought to be somewhat insufficient. The measurement location was also specified in 2.4 validation using real data. thank you
Point 6 : The Description is unclear in Table 1. It needs further explanation
Response 6: The following additional explanation has been added to “2.3 Development an algorithm for flexible force mat weight estimation.”; “There are a total of 6 features used in model learning. “Sensor value” is Total values ​​of data in activated cells. "No.Cell" means the number of sensors activated during weighting, and "CoP row" & "CoP column" are values ​​calculated using formulas 1 to 3. “Row Gap” & “Column Gap” refers to the width that is activated in the top, bottom, left, and right directions.
Point 7 : The curves in Figure 5 should be explained in details.
Response 7 : Figure 5 (now figure 7) is a result graph showing changes at each measurement location even if the weight is the same. Opinions on the curve were presented in discussion. Please understand that further explanation is difficult.
Point 8 : I suggest authors introduce and cite some excellent work bout flexible sensor system, such as Cell Reports Physical Science 2023, 4:101191.
Response 8 : I looked at the prior research you suggested and added it as a reference for this study.
Point 9 : Consider the interest of mentioning potential real-world applications of the force measurement mat.
Response 9 : In fact, as a follow-up study, we are planning a study to confirm accuracy based on dynamic data from real people. Once the results are obtained, I will report on the follow-up study. Thank you for your good comments.
Round 2
Reviewer 1 Report
Comments and Suggestions for Authors
The authors have improved the the paper considering the reviewer comment. No further suggestions.
Author Response
Point 1: The authors have improved the the paper considering the reviewer comment. No further suggestions.
Response 1: Thank you for your helpful comment. The quality of the manuscript seems to have improved. Thank you again.

Reviewer 2 Report
Comments and Suggestions for Authors
1. The author should also focus on the sensor technologies and its error factors, illustrating the working mechanisms of the sensor.
2. The author should accurately cite the references. some errors should be corrected such as reference 1 journal name is Cell Rep. Phys. Sci.; reference 5 should list the Authors; reference 17 journal name should be abbreviated.
Comments on the Quality of English LanguageMinor reversion
Author Response
Thank you for your review comments on this manuscript.
Point 1: The author should also focus on the sensor technologies and its error factors, illustrating the working mechanisms of the sensor
Response 1: The cause of the error factor is a problem with the communication method rather than the sensor unit, and this is presented in the introduction. Also, regarding the working mechanisms of the sensor, see “2.1. It was added as a sentence to “Introduction of flexible force measurement mat” and no other figures were inserted. A more detailed explanation is needed, but please understand that it is difficult to present all hardware. Thank you.
Point 2: The author should accurately cite the references. some errors should be corrected such as reference 1 journal name is Cell Rep. Phys. Sci.; reference 5 should list the Authors; reference 17 journal name should be abbreviated.
Response 2: A reference notation error has been corrected. thank you.
